# Practical Utility of a Clinical Pathway for Older Patients with Aspiration Pneumonia: A Single-Center Retrospective Observational Study

**DOI:** 10.3390/jcm13010230

**Published:** 2023-12-30

**Authors:** Taisuke Araki, Yoshitaka Yamazaki, Masanobu Kimoto, Norihiko Goto, Yuichi Ikuyama, Yuko Takahashi, Makoto Kosaka

**Affiliations:** 1First Department of Internal Medicine, Shinshu University School of Medicine, Matsumoto 390-8621, Japan; goto510@shinshu-u.ac.jp (N.G.); ikuyama@shinshu-u.ac.jp (Y.I.); 2Center of Infectious Diseases, Nagano Prefectural Shinshu Medical Center, Suzaka 382-8577, Japan; yoshitakayamazaki@hotmail.com (Y.Y.); masa017586@yahoo.co.jp (M.K.); m.kosaka@hotmail.co.jp (M.K.); 3Division of Clinical Laboratory, Nagano Prefectural Shinshu Medical Center, Suzaka 382-8577, Japan; takahashi-yuko-r@pref-nagano-hosp.jp

**Keywords:** aspiration pneumonia, clinical pathway, propensity score, propensity score matching, inverse probability of treatment weighting, older patients, prognosis

## Abstract

**Introduction**: Clinical pathways (CPWs) are patient management tools based on a standardized treatment plan aimed at improving quality of care. This study aimed to investigate whether CPW-guided treatment has a favorable impact on the outcomes of hospitalized older patients with aspiration pneumonia. **Method**: This retrospective study included patients with aspiration pneumonia, aged ≥ 65 years, and hospitalized at a community hospital in Japan. CPW implementation was arbitrarily determined by the attending physician upon admission. Outcomes were compared according to with or without the CPW (CPW-group and non-CPW groups). Propensity score (PS)-based analyses were used to control for confounding factors. Logistic regression analyses were conducted to evaluate the impact of CPW on the clinical course and outcomes. **Results**: Of 596 included patients, 167 (28%) received the CPW-guided treatment. The mortality rate was 16.4%. In multivariable model, CPW implementation did not increase the risk for total and 30-day mortality, and resulted in shorter antibiotic therapy duration (≤9 days) (PS matching (PSM): odds ratio (OR) 0.50, *p* = 0.001; inverse provability of treatment weighting (IPTW): OR 0.48, *p* < 0.001) and length of hospital stay (≤21 days) (PSM: OR 0.67, *p* = 0.05; IPTW: OR 0.66, *p* = 0.03). **Conclusions**: This study support CPW utility in this population.

## 1. Introduction

Aspiration pneumonia accounts for 5–15% of community-acquired pneumonia cases and is prevalent in patients with dysphagia due to underlying diseases and aging [1]. It is particularly frequent in older people and has poorer prognosis compared with non-aspiration pneumonia [2,3,4]. Despite advances in pneumonia management, including antibiotic therapy and vaccination, the prognosis of aspiration pneumonia remains unsatisfactory. Therefore, with the rapid aging of the population, aspiration pneumonia is becoming increasingly important in practice.

In Japan, which has the fastest-aging population worldwide, aspiration pneumonia is a primary concern in the medical care of older patients and imposes a significant socioeconomic burden. More than 60% of pneumonia cases occur in patients aged ≥ 75 years, and aspiration pneumonia is diagnosed in >70% of these cases [5]. Most patients with aspiration pneumonia are treated primarily in acute care beds, incurring enormous inpatient medical costs [6]. This unique situation in Japan underscores the need for better management of older patients with this disease.

A clinical pathway (CPW) is a quality management tool in medical care that helps standardize the care process; it enhances the quality of care with multifaceted benefits, such as improved patient outcomes, reduced burden on medical staff, and savings in healthcare costs [7]. Several studies investigating the effectiveness of CPWs in hospitalized patients with pneumonia have demonstrated that CPW implementation improved patient outcomes by providing quick and appropriate antibiotic treatment [8,9,10,11]. However, no studies have examined the practical feasibility of CPWs for older patients with aspiration pneumonia.

We hypothesized that CPWs could be applicable in the treatment of older patients with aspiration pneumonia and developed a CPW for our institution. In the present study, we investigate the impact of CPW-guided care on patient prognosis and clinical outcomes in this population.

## 2. Materials and Methods

### 2.1. Study Design and Patients

This was a retrospective analysis that included patients aged ≥ 65 years who were hospitalized due to aspiration pneumonia at an acute-phase community hospital with 300 beds in a rural city in Japan between April 2014 and March 2016. Participants were divided into two groups according to the type of treatment received. Patients who were applied and treated with the CPW were defined as the “CPW group”, while those who were not as the “non-CPW group”.

### 2.2. Diagnosis and Treatment

The diagnosis of aspiration pneumonia was based on clinical symptoms indicating pneumonia (i.e., fever, cough, and sputum), findings of infiltrates on chest radiography or computed tomography, and findings of aspiration as assessed by a trained speech therapist and nurse on admission. In our institution, the usual evaluation for aspiration by them included water and dry swallowing tests in patients without consciousness disorder, rarely using a swallowing video fluorography.

Disease severity was assessed according to the A-DROP scoring system proposed by the Japanese Respiratory Society (Appendix A) [12]. Evaluated parameters include age, dehydration, respiration, orientation, and blood pressure, with each assigned a score of 0 or 1. Based on the total score, disease severity is classified as mild (0 points), moderate (1–2 points), severe (3 points), or very severe (4–5 points).

Sputum samples were collected on admission whenever possible to determine the causative pathogens. Methicillin-resistant *Staphylococcus aureus*, *Pseudomonas aeruginosa*, *Acinetobacter baumannii*, *Stenotrophomonas maltophilia*, and extended-spectrum β-lactamase-producing Gram-negative bacilli were defined as potential drug-resistant (PDR) pathogens.

The CPW was constructed according to previously proposed operational definitions [7]; the design and components are shown in Figure 1. CPW implementation was determined arbitrarily by the attending physician on admission. Patients in the CPW group received 1000 mL of infusion and 1.5 g × 3 per day (every 8 h) of ampicillin/sulbactam (ABPC/SBT) as an initial antibiotic agent from hospital days 1 to 8. Blood tests and chest radiography were performed on hospital days 4 and 8 to assess treatment efficacy. The CPW also encompassed comprehensive care programs regarding nursing care, oral hygiene, nutrition support, multidisciplinary rehabilitation, and hospital discharge support. The attending physician could modify all interventions in the CPW as needed. Patients in the non-CPW group received treatment at the discretion of their attending physician. Antibiotic regimens, testing for efficacy assessment, and timing of multidisciplinary care interventions were determined by an attending physician.

### 2.3. Data Collection

Demographic and clinical data were extracted from the hospital electronic medical records and included age, sex, residence location, comorbidities, laboratory test results, antibiotic therapy (first-choice agent, treatment duration, and antibiotic change), length of hospital stay (LOS), and survival outcomes (in-hospital total and 30-day mortality). A change in the initial antibiotic regimen was defined as an initial treatment failure.

### 2.4. Study Outcomes

To evaluate the impact of CPW-guided treatment on patients’ clinical course and outcome, the association between CPW implementation and the following four factors was assessed as the study outcome: (1) total mortality, (2) 30-day mortality, (3) prolonged antibiotic therapy duration, and (4) prolonged LOS. As for (3) and (4), periods greater than the median value were defined as “prolonged”.

### 2.5. Statistical Analyses

Continuous data were summarized as median values with interquartile ranges or mean values with standard deviations based on their distribution, and categorical data were summarized as frequencies with percentages. The correlation between antibiotic therapy duration and LOS was tested by Spearman’s rank correlation coefficient.

As a retrospective nature of this study, there were no prespecified inclusion criteria for the implementation of CPW. Therefore, we assumed that baseline patient status and clinical information upon admission would potentially be confounding in implementing CPW. We performed propensity score (PS)-based analyses to control for biases related to patient background and potential confounding factors affecting CPW implementation. The PS was calculated by logistic regression analysis with the following factors potentially associated with survival outcomes as covariates: age, sex, residence location, body mass index, A-DROP score, serum albumin level, PDR pathogen detection, and history of malignancy and chronic heart failure. Two hypothetical cohorts were generated for further outcome analysis, one based on PS matching (PSM) and one based on inverse probability of treatment weighting (IPTW). Because PSM cannot estimate treatment effects in unmatched cases, the IPTW method was also employed to complement it. Differences between the CPW and non-CPW groups were presented as standardized mean differences (SMDs). Variables with SMD less than 0.1 were considered well-balanced between the two groups. PSM was conducted in a 2:1 ratio using a standardized deviation width of 0.20 for the logit transformation of the estimated PS. In the IPTW cohort, trimmed weights with a PS of less than the first percentile and greater than the ninety-ninth percentile were used to minimize extreme weighting influences. Logistic regression analyses were conducted to estimate the odds ratios (ORs) for the impact of CPW implementation on study outcomes with corresponding 95% confidence intervals (CIs). 

All statistical analyses were performed using EZR software (version 1.60) (Saitama Medical Center, Jichi Medical University, Saitama, Japan). All tests were two-sided, and statistical significance was set at *p* < 0.05 [13].

## 3. Results

### 3.1. Patients’ Characteristics

Overall, 596 patients were included in this analysis (Figure 2), 429 (72%) in the non-CPW group and 167 (28%) in the CPW group. Patients’ characteristics are presented in Table 1. 

In the crude cohort, no marked difference was observed between the two groups. However, compared with the non-CPW group, the CPW group had a lower proportion of patients with severe disease (48.7% vs. 39.5%; SMD = 0.19), a higher proportion of patients initially treated with ABPC/SBT (80.9% vs. 96.4%; SMD = 0.51), and fewer subsequent antibiotic changes (22.4% vs. 10.2%; SMD = 0.34). A sputum test was conducted in 88.9% of patients. Although the prevalence of PDR pathogens was slightly lower in the CPW than in the non-CPW group (18.0% vs. 24.7%; SMD = 0.17), microbiological results showed no significant differences between the groups (Appendix A). The median antibiotic therapy duration and LOS were 9.0 and 21.0 days, respectively, both shorter in the CPW than in the non-CPW group. We observed a positive correlation between antibiotic therapy duration and LOS (*r* = 0.51, *p* < 0.001; Appendix A). The overall mortality rate was 16.4%; total mortality was slightly lower in the CPW than in the non-CPW group (13.2% vs. 17.7%; SMD = 0.13), while 30-day mortality was comparable (8.4% vs. 9.1%; SMD = 0.03). 

Both in the PSM and IPTW cohorts, variations in background factors including the disease severity were well adjusted for. Similarly to what was observed in the crude cohort, compared with the non-CPW group, the CPW group had a higher proportion of patients with ABPC/SBT as the first-choice antibiotic therapy (PSM cohort: 80.4% vs. 96.8%; SMD = 0.54, IPTW cohort: 80.7% vs. 96.9%: SMD = 0.53), fewer treatment failures(PSM cohort: 80.4% vs. 96.8%; SMD = 0.54, IPTW cohort: 80.7% vs. 96.9%: SMD = 0.53), and shorter antibiotic therapy duration (PSM cohort: SMD = 0.25, IPTW cohort: SMD = 0.27) and LOS (PSM cohort: SMD = 0.15, IPTW cohort: SMD = 0.15).

### 3.2. Outcome Analysis

Multivariable logistic regression analysis in the crude cohort revealed that low albumin levels (OR, 2.43; 95% CI, 1.45–4.05; *p* < 0.001), low body mass index (OR, 4.47; 95% CI, 2.59–7.74; *p* < 0.001), and severe disease (A-DROP score: 3–5 points; OR, 2.13; 95% CI, 1.32–3.43; *p* = 0.002) were significantly associated with mortality events, whereas the use of CPW was not (OR, 0.75; 95% CI, 0.43–1.30; *p* = 0.30; Appendix A). 

The results of the univariable and multivariable logistic regression analyses for the association between CPW implementation and mortality events are presented in Table 2. First-choice ABPC/SBT and initial treatment failure were entered as explanatory variables in the multivariable model. In all models analyzed in the unadjusted and adjusted cohorts, CPW implementation did not increase mortality risks, neither for total (PSM cohort: OR, 0.96; 95% CI, 0.54–1.72. IPTW cohort: OR, 1.00; 95% CI, 0.58–1.74) nor for 30-day mortality (PSM cohort: OR, 1.07; 95% CI, 0.52–2.20. IPTW cohort: OR, 1.12; 95% CI, 0.56–2.22). 

The results of the univariable and multivariable analyses for the association between CPW implementation and antibiotic therapy duration and LOS are presented in Table 3. In these models, antibiotic therapy duration of ≥9 days and LOS of ≥21 days were the objective variables. First-choice ABPC/SBT and mortality events were entered as explanatory variables in the multivariable model. In all models, CPW implementation reduced the risks for prolonged antibiotic therapy duration (PSM cohort: OR, 0.50; 95% CI, 0.33–0.77; *p* = 0.001. IPTW cohort: OR, 0.48; 95% CI, 0.32–0.71; *p* < 0.001) and prolonged LOS (PSM cohort: OR, 0.67; 95% CI, 0.45–0.99; *p* = 0.05. IPTW cohort: OR, 0.66; 95% CI, 0.45–0.97; *p* = 0.03).

## 4. Discussion

This study examined the clinical utility of CPW-guided treatment for hospitalized older patients with aspiration pneumonia. The total mortality rate in the study population was 16.4%, which is comparable to the historical data on pneumonia in older adults in Japan [2,4,14,15,16]. After adjusting for background factors, we observed that CPW-guided treatment did not increase mortality risks and was associated with shorter antibiotic therapy duration and LOS compared with those in the non-CPW group. To the best of our knowledge, this is the first study to investigate the impact of CPW-guided treatment on the clinical outcomes of hospitalized older patients with aspiration pneumonia. The study findings may potentially advocate a new approach for the management of this patient population.

The most notable finding of this study was that CPW-guided treatment did not increase mortality risks in older patients with aspiration pneumonia. The total and 30-day mortality rates did not significantly differ between the non-CPW and CPW groups. Several prior studies on community-acquired pneumonia have reported that implementing a CPW focused on guideline-concordant antibiotic regimens reduced mortality risks [8,9,10]. The authors opined that providing prompt and optimal antibiotic therapy contributed to risk reduction. However, these studies included relatively young patients, and this difference in the study population may explain the discrepancy in results with our study, which included only older patients. Namely, the prognosis of aspiration pneumonia in older patients is reportedly affected by patient-specific factors, such as nutritional status [17,18,19] and underlying diseases [20], and not by antibiotic therapy [21]. Indeed, in the current study, lower albumin levels, lower body mass index, and disease severity were independent poor prognostic factors in the crude cohort. Importantly, no significant association between CPW use and mortality risks was observed, even after adjusting for confounding factors that could potentially affect the CPW implementation, including disease severity. These findings indicate that our CPW would not affect the survival outcomes in this population.

We also found that CPW implementation was associated with a shorter antibiotic therapy duration. Furthermore, more patients received ABPC/SBT as a first-choice agent, and fewer antibiotic changes due to subsequent treatment failures were observed in the CPW than in the non-CPW group. Notably, treatment failures were not related to the isolation of PDRs, which ABPC/SBT could not cover; thus, the findings of the present study support the limited necessity of broad-spectrum antibiotics to target PDR pathogens in older patients with pneumonia proposed in recent years. These findings also suggest that the ABPC/SBT regimen specified in our CPW (4.5 g per day for 8 days) was potentially effective for empirical treatment of aspiration pneumonia in the current study setting. However, an antibiotic regimen should be determined considering the patient’s background and local antibiogram. ABPC/SBT is commonly recommended as an initial antibiotic for community-acquired pneumonia [22,23], for which the optimal regimen to be incorporated into a CPW requires further investigation.

A reduction in the LOS potentially contributes to reduced hospital care costs and medical staff burden and is therefore one of the critical indicators of CPW effectiveness. In the current study, CPW implementation was associated with a shorter LOS, although mortality rates were comparable in the two groups. This finding suggests that CPW implementation may have promoted earlier discharge in patients who survived, which can be due to several reasons. First, as a positive correlation between antibiotic therapy duration and LOS was observed, the reduced antibiotic therapy duration in the CPW group might have impacted the LOS. Second, the multidisciplinary intervention prescribed in the CPW group may have facilitated early discharge, resulting in shortened LOS. As demonstrated in prior studies on the effectiveness of multidisciplinary interventions in older inpatients [24], the CPW-guided intervention program may have promoted earlier recovery and discharge in our cohort. This suggests the utility of the CPW in the management of older patients with aspiration pneumonia. However, the reduction in LOS was not translated into a reduced risk for mortality in the CPW group, which emphasizes the need for different approaches to improving patient survival outcomes.

Our CPW was unique in that it not only specified the antibiotic regimen but also the intervention by multidisciplinary staff in the care process from admission. This design offered a strength in our study setting, which has a different sociomedical background from that in Western countries. In Japan, long-term care or residential beds for the older population are insufficient; hence, many older patients with pneumonia who have impaired daily activity abilities or require extended care receive convalescent rehabilitation following antibiotic therapy in acute care beds. The reported LOS for pneumonia in Japan is as long as 29 days [6], much longer than that reported in Western countries. Therefore, when considering that our CPW included a series of interventions, such as comprehensive rehabilitation and discharge support programs, following the acute phase treatment protocol is rational. Furthermore, facilitating multidisciplinary interventions from admission enables the quick sharing of patient information and practical social support in anticipation of discharge.

On the other hand, the external validation of the design and utility of our CPW for older patients with aspiration pneumonia might be difficult. It should be noted that the applicability of our findings in all countries, regions, and clinical settings remains uncertain. Factors such as geography, socio-medical economics, distribution of patients per facility, and manpower of medical staff, including physicians, must be taken into account when designing an optimal CPW. The performance of a CPW would be maximized by modifying each setting for all pathway interventions. Importantly, the effectiveness of the CPW in each setting needs to be validated individually, as in the present study. Because the evidence regarding CPW for aspiration pneumonia in older patients is critically lacking, individual validation might make CPW feasible for implementation in this population.

The present study had several limitations. First, the diagnostic criteria of aspiration pneumonia in this study were defined independently. Clinical findings suspicious of pneumonia and confirmation of aspiration by visual inspection and swallowing test were incorporated into this study’s diagnosis of aspiration pneumonia. The lack of established diagnostic criteria may have created a potential selection bias in patient inclusion. Second, no strict inclusion criteria were defined for the CPW implementation; decisions were based on the discretion of the attending physicians. Indeed, in the crude cohort, the proportion of patients with severe disease was higher in the non-CPW group. Although confounding, including disease severity, was addressed as much as possible in the PS-based analysis, other potential biases in CPW application may have been presented. Furthermore, data on concomitant medications, such as antipsychotics and proton pump inhibitors which are reported as a risk factor for aspiration pneumonia, were not collected in this study. These factors may have been residual confounders in this study. Third, as described above, the CPW component is unique to our study setting and does not have sufficient external validity yet. For example, our CPW employed ABPC/SBT 4.5 g per day as the antibiotic regimen, which is less than the standard dosage of ABPC/SBT for adults in Japan (6 g per day). Although the optimal dosage of ABPC/SBT in pneumonia in the older patients included in this study has not been established, we determined the dosage for CPW based on the assumption that participants have a smaller body weight and body surface area compared to younger adult patients. As a result, the mortality rate in the CPW group was 13.2%, indicating that this regimen was not excessively inappropriate. The optimal antibiotic regimen for aspiration pneumonia in older patients remains controversial and requires further validation. Furthermore, the multidisciplinary interventions prescribed by our CPW also require an optimized design that reflects the situation within the facility and the socio-medical background in the local area. Therefore, our CPW is not directly applicable in other regions or facilities. Forth, the study outcomes we employed in this study may be insufficient to evaluate the essential efficacy of CPW on aspiration pneumonia in older patients. The present study failed to examine the potential benefits of CPW, such as reduction in medical staff burden and healthcare costs, long-term prognosis of participants, and prevention of recurrent pneumonia. However, our study suggests that CPW in this study setting may be useful not only in therapeutic processes such as antibiotics but also in multidisciplinary interventions, which we hope will be further validated in external settings.

## 5. Conclusions

CPW-guided treatment favorably impacted antibiotic therapy duration and LOS without affecting mortality risks. We believe that CPW implementation facilitated smooth multidisciplinary intervention, positively impacting clinical outcomes. Our findings suggest the preliminary utility of our CPW for the management of older patients with aspiration pneumonia, providing a foothold for future clinical expansion.

## Figures and Tables

**Figure 1 jcm-13-00230-f001:**
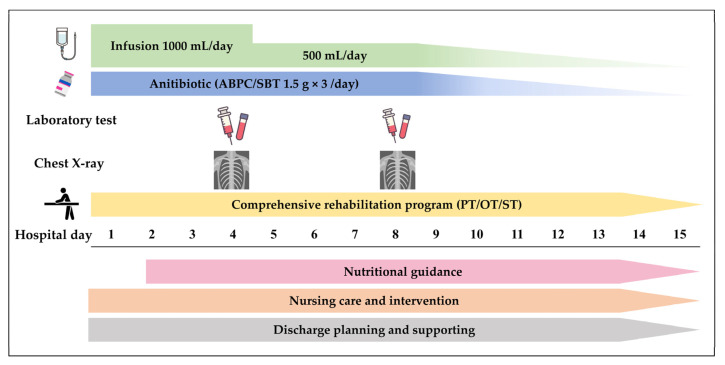
Schematic diagram and timeline of the clinical pathway used for older patients with aspiration pneumonia in this study. Abbreviations: OT, occupational therapist; PT, physical therapist; ST, speech therapist. Note: All interventions could be modified at the discretion of the attending physician.

**Figure 2 jcm-13-00230-f002:**
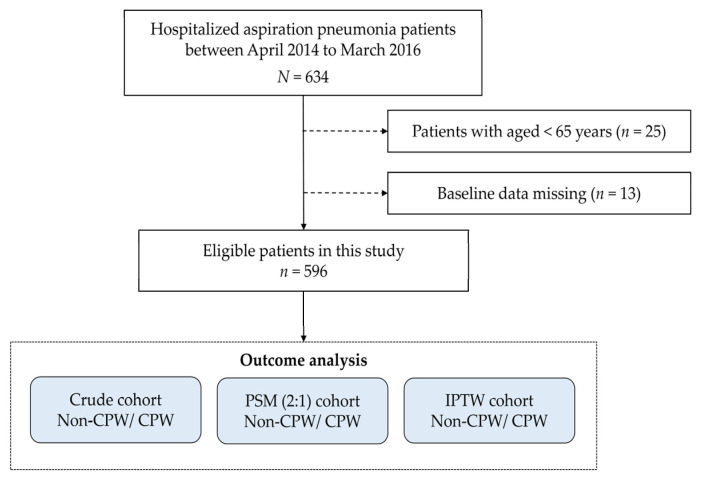
Study flowchart. Abbreviations: CPW, clinical pathway; IPTW, inverse probability of treatment weighting; PSM, propensity score matching.

**Table 1 jcm-13-00230-t001:** Patient characteristics in the study cohorts.

	Crude Cohort			PSM Cohort ^a^			IPTW Cohort ^a^		
Characteristics, *n* (%)	Non-CPW *n* = 429	CPW *n* = 167	SMD	Non-CPW *n* = 316	CPW *n* = 158	SMD	Non-CPW %	CPW %	SMD
**Age, years, median (range)**	87 (65, 104)	86 (65, 104)	0.14	86 (65, 104)	86 (65, 104)	0.04	87 (65, 104)	87 (65, 104)	0.01
**Sex**									
Male	257 (59.9)	107 (64.1)	0.09	193 (61.1)	100 (63.3)	0.05	61.2	61.8	0.01
Female	172 (40.1)	60 (35.9)		123 (38.9)	58 (36.7)		38.8	38.2	
**Body mass index, kg/m^2^**	18.11 ± 3.58	18.28 ± 3.76	0.05	18.32 ± 3.73	18.26 ± 3.75	0.02	18.19 ± 3.61	18.11 ± 3.73	0.02
**Resident location**									
Own home	205 (47.8)	88 (52.69)	0.10	167 (52.8)	85 (53.8)	0.02	49.3	50.2	0.02
Other	224 (52.2)	79 (47.3)		149 (47.2)	73 (46.2)		50.7	49.8	
**Comorbidity**									
Dementia	202 (47.1)	84 (50.3)	0.06	148 (46.8)	80 (50.6)	0.08	46.8	51.5	0.09
Stroke sequelae	162 (37.8)	57 (34.1)	0.08	121 (38.3)	53 (33.5)	0.10	38.2	33.1	0.11
Diabetes mellitus	80 (18.5)	29 (17.4)	0.03	55 (17.4)	29 (18.4)	0.03	18.9	18.2	0.02
Chronic heart failure	73 (17.0)	21 (12.6)	0.13	42 (13.3)	20 (12.7)	0.02	15.7	14.8	0.02
Malignancy	60 (14.0)	23 (13.8)	0.006	44 (13.9)	22 (13.9)	<0.001	14.1	14.9	0.02
Neurodegenerative disease	14 (3.3)	4 (2.4)	0.05	8 (2.5)	4 (2.5)	<0.001	3.2	2.1	0.07
**Laboratory test results**									
White blood cells, cells/µL	94 (68, 130)	91 (72, 123)	0.04	97 (68, 129)	93 (72, 123)	0.009	95 (68, 130)	90 (72, 122)	0.04
C-reactive protein, mg/dL	5.92 (2.00, 11.93)	6.15 (2.23, 12.44)	0.12	6.07 (2.15, 12.50)	5.94 (2.08, 12,82)	0.07	5.93 (2.00, 11.94)	6.03 (2.11, 12.41)	0.09
Albumin, g/dL	3.0 ± 0.6	3.1 ± 0.6	0.12	3.0 ± 0.6	3.1 ± 0.6	0.11	3.0 ± 0.6	3.1 ± 0.6	0.06
Blood urea nitrogen, mg/dL	19.3 (14.7, 26.4)	18.9 (14.9, 27.7)	0.02	18.4 (14.2, 25.3)	19.0 (15.4, 27.9)	0.13	19.0 (14.5, 25.8)	19.0 (15.1, 28.0)	0.07
Creatinine, mg/dL	0.72 (0.52, 1.02)	0.73 (0.56, 0.96)	0.06	0.72 (0.52, 1.01)	0.76 (0.56, 1.01)	0.004	0.73 (0.52, 1.02)	0.73 (0.55, 0.99)	0.05
PDR pathogen detection	106 (24.7)	30 (18.0)	0.17	52 (16.5)	30 (19.0)	0.07	22.8	22.8	0.002
**A-DROP score**									
0–2, points	220 (51.3)	101 (60.5)	0.19	180 (57.0)	92 (58.2)	0.03	53.9	54.5	0.01
3–5, points	209 (48.7)	66 (39.5)		136 (43.0)	66 (41.8)		46.1	45.5	
**Clinical course and outcome**									
First choice ABPC/SBT	347 (80.9)	161 (96.4)	0.51	254 (80.4)	153 (96.8)	0.54	80.7	96.9	0.53
Initial treatment failure	96 (22.4)	17 (10.2)	0.34	66 (20.9)	17 (10.8)	0.28	22.3	10.4	0.33
Administration period, days	10.0 (7.0, 14.0)	9.0 (7.0, 10.0)	0.27	9.0 (7.0, 14.0)	9.0 (7.0, 10.0)	0.25	9.6 (7.0, 14.0)	9.0 (7.0, 10.0)	0.27
Length of hospital stay, days	22.0 (15.0, 38.0)	18.0 (13.0, 31.5)	0.19	22.0 (14.8, 37.3)	18.0 (13.0, 32.3)	0.15	22 (14, 38)	18 (13, 33)	0.15
Mortality event	76 (17.7)	22 (13.2)	0.13	52 (16.5)	21 (13.3)	0.09	17.3	14.9	0.07
30-day mortality event	39 (9.1)	14 (8.4)	0.03	26 (8.2)	13 (8.2)	<0.001	8.8	9.9	0.04

^a^ Adjusted for age, sex, residence location, body mass index, A-DROP score, serum albumin level, PDR pathogen detection, and history of malignancy and chronic heart failure. Abbreviations: ABPC/SBT, ampicillin/sulbactam; CPW, clinical pathway; IPTW, inverse probability of treatment weighting; PDR, potentially drug resistant; PSM, propensity score matching; SMD, standardized mean difference.

**Table 2 jcm-13-00230-t002:** Association of clinical pathway use and mortality event before and after adjustment.

	Odds Ratio	95% Confidence Interval	*p*
**Total mortality event**			
**Univariable model**			
Crude cohort	0.71	0.42–1.18	0.18
PSM cohort	0.80	0.46–1.38	0.42
IPTW cohort	0.80	0.47–1.37	0.42
**Multivariable model**			
PSM cohort	0.96	0.54–1.72	0.89
IPTW cohort	1.00	0.58–1.74	0.99
**30-day mortality event**			
**Univariable model**			
Crude cohort	0.92	0.48–1.73	0.79
PSM cohort	1.04	0.52–2.10	0.91
IPTW cohort	1.01	0.52–1.98	0.97
**Multivariable model**			
PSM cohort	1.07	0.52–2.20	0.85
IPTW cohort	1.12	0.56–2.22	0.76

Abbreviations: IPTW, inverse probability of treatment weighting; PSM, propensity score matching.

**Table 3 jcm-13-00230-t003:** Association of clinical pathway use and antibiotic therapy duration and length of hospital stay before and after adjustment.

	Odds Ratio	95% Confidence Interval	*p*
**Prolonged antibiotic therapy duration** **(≥9 days)**			
**Univariable model**			
Crude cohort	0.44	0.30–0.64	<0.001
PSM cohort	0.45	0.30–0.68	<0.001
IPTW cohort	0.43	0.29–0.64	<0.001
**Multivariable model**			
PSM cohort	0.50	0.33–0.77	0.001
IPTW cohort	0.48	0.32–0.71	<0.001
**Prolonged length of hospital stay** **(≥21 days)**			
**Univariable model**			
Crude cohort	0.59	0.41–0.85	0.005
PSM cohort	0.64	0.43–0.94	0.02
IPTW cohort	0.65	0.45–0.95	0.02
**Multivariable model**			
PSM cohort	0.67	0.45–0.99	0.05
IPTW cohort	0.66	0.45–0.97	0.03

Abbreviations: IPTW, inverse probability of treatment weighting; PSM, propensity score matching.

## Data Availability

The data presented in this study are available on request from the corresponding author.

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
