# Peer review of "Practical Utility of a Clinical Pathway for Older Patients with Aspiration Pneumonia: A Single-Center Retrospective Observational Study"

_jcm, 2023, doi:10.3390/jcm13010230_

Round 1

Reviewer 1 Report

Comments and Suggestions for Authors

Title: Practical Utility of a Clinical Pathway for Older Patients with Aspiration Pneumonia: A Single Centre Retrospective Observational Study

The title of the manuscript was interesting because it relates with high morbidity and mortality according to the treatment. Eventhough, there are several things that need to be clarified or revised as follows:

  1. The study design was retrospective-observational study and older patient with aspiration pneumonia as population sample. The gold standard for diagnosis the aspiration pneumonia needs to be explained. Is the diagnosis enough only based on clinical symptoms, findings of infiltrates on chest radiography or computed tomography, and findings of aspiration as assessed by a trained speech therapist and nurse on admission indicating aspiration pneumonia?
  2. There is a usual care group besides the CPW (Clinical Pathway). The usual care is need to be clarified clearly what the operational definition is.
  3. Please give the information about the number of ethical clearance.
  4. The use of antibiotic (APBC/SBT 15x3 / day) as one of some components in CPW is unwarranted, it should be data which backs up this choice of antibiotics including the daily dose.
  5. In lines 258-259, there is sentences “Additionally, the indication for implementing…”. What is meaning? Is the use of CPW not adhered by physician who use CPW? There still the physician who don’t acknowledge the use of CPW. It should be strictly implemented by the attending discretion physician. This creates bias so that the conclusions become unreliable.

Author Response

To the Reviewer 1

Thank you for spending your time reviewing our manuscript. We fully agree with your comments and have revised the text and tables/figures. Please kindly see our response below and the relevant sections in the manuscript.

Comment 1

The gold standard for diagnosis the aspiration pneumonia needs to be explained. Is the diagnosis enough only based on clinical symptoms, findings of infiltrates on chest radiography or computed tomography, and findings of aspiration as assessed by a trained speech therapist and nurse on admission indicating aspiration pneumonia?

  • As you mentioned, the diagnosis of aspiration pneumonia in the elderly is encompassed by heterogeneity; no gold standard for diagnostic criteria has been established. We sincerely understand that the diagnostic criteria used in this study are incomplete. In our institution, in patients without impaired consciousness, aspiration is diagnosed using the water or dry swallowing test and rarely using a swallowing video fluorography (added to the Material and Methods; L74~76). The retrospective nature of this study made it difficult to adapt detailed criteria for the diagnosis of aspiration pneumonia, and data on swallowing tests were unavailable. These are critical limitations of this study and have been added to the Discussion.

Comment 2

There is a usual care group besides the CPW (Clinical Pathway). The usual care is need to be clarified clearly what the operational definition is.

  • In this study, patients to whom the CPW was not applied were categorized as the “usual care group”. In the usual care group, all interventions, including antibiotic therapy, were prescribed at the discretion of an attending physician. No strict operational criteria were defined for the usual care group. Because the name, “usual care”, might be confusing, we changed the term “usual care group” to “non-CPW group”. Please see the changes in the relevant places in the manuscript.

Comment 3

Please give the information about the number of ethical clearance.

  • Information on ethical clearance is included in the manuscript at L331~336.

Comment 4

The use of antibiotic (APBC/SBT 15x3 / day) as one of some components in CPW is unwarranted, it should be data which backs up this choice of antibiotics including the daily dose.

  • As you pointed out, the 1.5g × 3/day of ABPC/SBT is not based on concrete evidence. We thought, based on the PK/PD theory, that three or four doses of ABPC/SBT would be more effective than two doses per day. This was demonstrated in another study by Suzuki et al* Given that the study population in our study is elderly with smaller physical characteristics [mean body weight, 43.7 ± 26.3kg; mean body surface area (du bois), 1.35 ± 0.18 m2], the current dosage was chosen; however, we do not have data that this regimen is optimal treating for aspiration pneumonia in the elderly patients. Instead, CPW antibiotics were allowed to be arbitrarily changed by the attending physician depending on the patient's condition. We fully understand that this point needs further external validation.

*Suzuki T, Sugiyama E, Nozawa K, et al. Effects of dosing frequency on the clinical efficacy of ampicillin/sulbactam in Japanese elderly patients with pneumonia: A single-center retrospective observational study. Pharmacol Res Perspect. 2021;9(2):e00746. doi:10.1002/prp2.746.

Comment 5

In lines 258-259, there is sentences “Additionally, the indication for implementing…”. What is meaning? Is the use of CPW not adhered by physician who use CPW? There still the physician who don’t acknowledge the use of CPW. It should be strictly implemented by the attending discretion physician. This creates bias so that the conclusions become unreliable.

  • We apologize for the incomprehensible description. The description implies a potential selection bias in applying the CPW, because of the retrospective nature of the present study.

Reviewer 2 Report

Comments and Suggestions for Authors

The main objectives of this study were to investigate the impact of Clinical Pathways (CPWs) on the outcomes of older patients (≥ 65 years) hospitalized with aspiration pneumonia at a community hospital in Japan. The study aimed to assess whether CPW-guided treatment, determined by attending physicians, had a favorable effect on clinical outcomes compared to usual care. The study utilized propensity score matching and inverse probability of treatment weighting to control for confounding factors and employed logistic regression analyses to evaluate the influence of CPW utilization on the clinical course and outcomes. The study found that CPW-guided treatment did not increase the risk of total and 30-day mortality, and it resulted in shorter durations of antibiotic therapy and hospital stay, suggesting the potential utility of CPWs in this patient population. Although the study presents a certain scientific interest, there are some concerns, and here are some important comments that will help readers understand the practical significance of these studies:

1. The criteria for diagnosing aspiration pneumonia, please, further elaboration on how treatments in the usual care group were determined could enhance understanding of the differences between the two groups.

2. The use of propensity score-based analyses to control for biases and confounding is a strong aspect of the study. However, a more detailed explanation of why these particular statistical methods were chosen and their appropriateness for the study design would be beneficial. Additionally, discussing the limitations of these methods and any assumptions made during the analysis would strengthen this section.

3. While the study provides valuable insights, it's important to discuss the generalizability of the findings, especially since it is a single-center study in a specific geographical location.

4. While the CPW is detailed, mentioning the need for external validation in different settings or populations would be useful, as the findings may not be directly applicable to other settings due to differences in healthcare systems, patient populations, etc.

5. The clear presentation of patient characteristics in both the CPW and usual care groups is commendable. However, the significant difference in the proportion of patients with severe disease between the two groups (48.7% vs. 39.5%) might introduce a confounding variable affecting the outcomes. It would be beneficial to discuss how this disparity might have influenced the study results.

6. The detailed reporting of treatment protocols and the correlation between antibiotic therapy duration and length of stay (LOS) are strengths of the study. The observation of shorter median antibiotic therapy duration and LOS in the CPW group compared to the usual care group is an interesting finding. However, the causative link between CPW implementation and these outcomes should be explored further, considering other potential influencing factors.

7. The finding that CPW implementation did not significantly affect mortality rates is crucial. However, it's important to consider and discuss the clinical relevance of this finding in the context of overall patient care and quality of life.

8. While the results are specific to the study's setting, discussing their generalizability to other populations or healthcare systems would be valuable. This could include speculating on how these findings might be replicated or differ in other contexts.

9. Addressing any limitations of the results, such as potential biases in patient selection or treatment assignment, would provide a more balanced view.

10. The discussion about the CPW not increasing mortality risks and its association with shorter antibiotic therapy duration and LOS is well-articulated. However, it would be beneficial to delve deeper into the potential mechanisms behind these observations. For instance, exploring how the CPW might have influenced these outcomes beyond the direct effects of treatment protocols could provide a richer understanding.

11. While you mention the need for external validation, a more detailed discussion on how the CPW could be adapted or tested in other settings would be useful.

Comments on the Quality of English Language

The paper required minor editing

Author Response

To the Reviewer 2

Thank you for spending your time reviewing our manuscript. We fully agree with your comments and have revised the text and tables/figures. Please kindly see our response below and the relevant sections in the manuscript.

Comment 1

The criteria for diagnosing aspiration pneumonia, please, further elaboration on how treatments in the usual care group were determined could enhance understanding of the differences between the two groups.

  • In this study, patients to whom the CPW was not applied were categorized as the “usual care group”. In the usual care group, all treatment plans were determined according to the discretion of the attending physician, and any interventions were not predefined. The term “usual care group” was changed to the “non-CPW group” because we felt it was misleading.

Comment 2

The use of propensity score-based analyses to control for biases and confounding is a strong aspect of the study. However, a more detailed explanation of why these particular statistical methods were chosen and their appropriateness for the study design would be beneficial. Additionally, discussing the limitations of these methods and any assumptions made during the analysis would strengthen this section.

  • A propensity score-based analysis was employed to adjust for confounding influences on CPW application. The IPTW method was used to complement the PS-matching, as it is difficult to estimate the effect of treatment on unmatched cases (as described in L121~138).

Comment 3

While the study provides valuable insights, it's important to discuss the generalizability of the findings, especially since it is a single-center study in a specific geographical location.

  • Thanks for your feedback. The issues you mentioned are critical limitations of this study, and we have added specific descriptions in the Discussion.

Comment 4

While the CPW is detailed, mentioning the need for external validation in different settings or populations would be useful, as the findings may not be directly applicable to other settings due to differences in healthcare systems, patient populations, etc.

  • Relating to your “Comment 3”, we will mention this point in Discussion as well.

Comment 5

The clear presentation of patient characteristics in both the CPW and usual care groups is commendable. However, the significant difference in the proportion of patients with severe disease between the two groups (48.7% vs. 39.5%) might introduce a confounding variable affecting the outcomes. It would be beneficial to discuss how this disparity might have influenced the study results.

  • As you pointed out, the disease severity was different between the CPW and usual care groups in the crude cohort. Indeed, it suggests that disease severity upon admission may have been a potential bias for attending physicians to hesitate in applying the CPW. Therefore, disease severity was incorporated into the adjustment factor for propensity scores and was well-balanced between the two groups in the PSM and IPTW cohorts; in the PSM and IPTW cohorts, variables with SMD < 0.1 were considered well-balanced between the two groups. We considered that the variation in disease severity in applying the CPW was adequately adjusted for in the outcome analyses in the PSM and IPTW cohorts.

Comment 6

The detailed reporting of treatment protocols and the correlation between antibiotic therapy duration and length of stay (LOS) are strengths of the study. The observation of shorter median antibiotic therapy duration and LOS in the CPW group compared to the usual care group is an interesting finding. However, the causative link between CPW implementation and these outcomes should be explored further, considering other potential influencing factors.

  • Thank you for your precise feedback. It is difficult to discuss your points further with the data we have.

Comment 7

The finding that CPW implementation did not significantly affect mortality rates is crucial. However, it's important to consider and discuss the clinical relevance of this finding in the context of overall patient care and quality of life.

  • It is difficult to discuss this point specifically with our data. We hope that in the future, data will be collected on geriatric assessment tools as essential outcomes of long-term CPW effectiveness. This point was described in the Discussion as a limitation of the study (L286~313).

Comment 8

While the results are specific to the study's setting, discussing their generalizability to other populations or healthcare systems would be valuable. This could include speculating on how these findings might be replicated or differ in other contexts.

  • Relating to your “Comment 3 and 4”, we will mention this point in Discussion as well.

Comment 9

Addressing any limitations of the results, such as potential biases in patient selection or treatment assignment, would provide a more balanced view.

  • Relating to your “Comment 3, 4 and 8”, we will mention this point in Discussion as well.

Comment 10

The discussion about the CPW not increasing mortality risks and its association with shorter antibiotic therapy duration and LOS is well-articulated. However, it would be beneficial to delve deeper into the potential mechanisms behind these observations. For instance, exploring how the CPW might have influenced these outcomes beyond the direct effects of treatment protocols could provide a richer understanding.

  • Thank you for your precise comments; it is possible that the implementation of CPW has improved communication within the medical team, resulting in shorter antibiotic therapy duration and LOS. However, this is not quantifiable data and lacks reproducibility. Therefore, we feel it is difficult to mention in the current analysis and discussion.

Comment 11

While you mention the need for external validation, a more detailed discussion on how the CPW could be adapted or tested in other settings would be useful.

  • Relating to your “Comment 3, 4, 8 and 9”, we will mention this point in Discussion as well.